# Ethnicity and Clinical Outcomes in Patients Hospitalized for COVID-19 in Spain: Results from the Multicenter SEMI-COVID-19 Registry

**DOI:** 10.3390/jcm11071949

**Published:** 2022-03-31

**Authors:** Jose-Manuel Ramos-Rincon, Lidia Cobos-Palacios, Almudena López-Sampalo, Michele Ricci, Manuel Rubio-Rivas, Francisco Martos-Pérez, Antonio Lalueza-Blanco, Sergio Moragón-Ledesma, Eva-María Fonseca-Aizpuru, Gema-María García-García, Jose-Luis Beato-Perez, Claudia Josa-Laorden, Francisco Arnalich-Fernández, Sonia Molinos-Castro, José-David Torres-Peña, Arturo Artero, Juan-Antonio Vargas-Núñez, Manuel Mendez-Bailon, Jose Loureiro-Amigo, María-Soledad Hernández-Garrido, Jorge Peris-García, Manuel-Lorenzo López-Reboiro, Bosco Barón-Franco, Jose-Manuel Casas-Rojo, Ricardo Gómez-Huelgas

**Affiliations:** 1Clinical Medicine Department, Miguel Hernandez University of Elche, 03550 Alicante, Spain; 2Internal Medicine Department, Regional University Hospital of Málaga, Biomedical Research Institute of Málaga (IBIMA), 29010 Malaga, Spain; cobospalacios@gmail.com (L.C.-P.); almu_540@hotmail.com (A.L.-S.); michele.ricci4@gmail.com (M.R.); ricardogomezhuelgas@hotmail.com (R.G.-H.); 3Medicine Department, University of Málaga, 29010 Malaga, Spain; 4Department of Internal Medicine, Bellvitge University Hospital, L’Hospitalet de Llobregat, 08907 Barcelona, Spain; mrubio@bellvitgehospital.cat; 5Internal Medicine Department, Costa del Sol Hospital, 29603 Marbella, Spain; pacomartos1@gmail.com; 6Internal Medicine Department, Doce de Octubre University Hospital, 28041 Madrid, Spain; lalueza@hotmail.com; 7Internal Medicine Department, Gregorio Marañón General University Hospital, 28007 Madrid, Spain; sergio.moragon@salud.madrid.org; 8Internal Medicine Department, Cabueñes University Hospital, 33394 Gijon, Spain; evamfonseca@yahoo.es; 9Internal Medicine Department, Badajoz University Hospital Complex, 06080 Badajoz, Spain; geminway21@hotmail.com; 10Internal Medicine Department, Albacete University Hospital Complex, 02006 Albacete, Spain; jlbeato@sescam.org; 11Internal Medicine Department, Royo Villanova Hospital, 50015 Zaragoza, Spain; claudiajosa@gmail.com; 12Internal Medicine Department, La Paz University Hospital, 28046 Madrid, Spain; farnalich@salud.madrid.org; 13Internal Medicine Department, Santiago de Compostela Clinic Hospital, 15706 Santiago de Compostela, Spain; sonia.molinos.castro@sergas.es; 14Lipids and Atherosclerosis Unit, Department of Internal Medicine, Maimonides Biomedical Research Institute of Cordoba (IMIBIC), Reina Sofia University Hospital, University of Cordoba, 14004 Cordoba, Spain; h42topej@uco.es; 15Spain CIBER Fisiopatologia de la Obesidad y la Nutricion, Carlos III Health Institute, 28029 Madrid, Spain; 16Internal Medicine Department, Doctor Peset University Hospital, 46017 Valencia, Spain; arturo.artero@uv.es; 17Internal Medicine Department, Puerta de Hierro University Hospital, Instituto de Investigación Sanitaria Puerta de Hierro—Segovia de Arana, 28222 Madrid, Spain; juanantonio.vargas@uam.es; 18Internal Medicine Department, San Carlos Clinco Hospital, 28040 Madrid, Spain; manuelmenba@hotmail.com; 19Internal Medicine Department, Moisès Broggi Hospital, 08970 Sant Joan Despí, Spain; jose.loureiro.amigo@gmail.com; 20Internal Medicine Department, Elda General University Hospital, 03600 Alicante, Spain; soleilhg@gmail.com; 21Internal Medicine Department, de Sant Joan d’Alacant University Clínic Hospital, 03550 Alicante, Spain; jorgeperisgarcia@gmail.com; 22Internal Medicine Department, Monforte de Lemos Hospital, 27400 Lugo, Spain; manuel.lorenzo.lopez.reboiro@sergas.es; 23Internal Medicine Department, University Hospital Virgen del Rocío, 41013 Sevilla, Spain; boscobar@icloud.com; 24Internal Medicine Department, Infanta Cristina University Hospital, Parla, 28981 Madrid, Spain; jm.casas@gmail.com

**Keywords:** COVID-19, SARS-CoV-2, ethnic groups, minority groups, migrants, Spain

## Abstract

(1) Background: This work aims to analyze clinical outcomes according to ethnic groups in patients hospitalized for COVID-19 in Spain. (2) Methods: This nationwide, retrospective, multicenter, observational study analyzed hospitalized patients with confirmed COVID-19 in 150 Spanish hospitals (SEMI-COVID-19 Registry) from 1 March 2020 to 31 December 2021. Clinical outcomes were assessed according to ethnicity (Latin Americans, Sub-Saharan Africans, Asians, North Africans, Europeans). The outcomes were in-hospital mortality (IHM), intensive care unit (ICU) admission, and the use of invasive mechanical ventilation (IMV). Associations between ethnic groups and clinical outcomes adjusted for patient characteristics and baseline Charlson Comorbidity Index values and wave were evaluated using logistic regression. (3) Results: Of 23,953 patients (median age 69.5 years, 42.9% women), 7.0% were Latin American, 1.2% were North African, 0.5% were Asian, 0.5% were Sub-Saharan African, and 89.7% were European. Ethnic minority patients were significantly younger than European patients (median (IQR) age 49.1 (40.5–58.9) to 57.1 (44.1–67.1) vs. 71.5 (59.5–81.4) years, *p* < 0.001). The unadjusted IHM was higher in European (21.6%) versus North African (11.4%), Asian (10.9%), Latin American (7.1%), and Sub-Saharan African (3.2%) patients. After further adjustment, the IHM was lower in Sub-Saharan African (OR 0.28 (0.10–0.79), *p* = 0.017) versus European patients, while ICU admission rates were higher in Latin American and North African versus European patients (OR (95%CI) 1.37 (1.17–1.60), *p* < 0.001) and (OR (95%CI) 1.74 (1.26–2.41), *p* < 0.001). Moreover, Latin American patients were 39% more likely than European patients to use IMV (OR (95%CI) 1.43 (1.21–1.71), *p* < 0.001). (4) Conclusion: The adjusted IHM was similar in all groups except for Sub-Saharan Africans, who had lower IHM. Latin American patients were admitted to the ICU and required IMV more often.

## 1. Introduction

There have been over 404 million reported COVID-19 infections and more than 5.7 million deaths worldwide in the COVID-19 pandemic as of 8 February 2022 [1]. In the United States of America, the pandemic has disproportionately affected racial and ethnic minority populations, who are at an increased risk of infection, hospitalization, and death [2,3,4]. Differences in access to healthcare and exposure risk may be driving higher infection and mortality rates among these groups [2,5,6]. In the UK, clear differences have been found in the risk of COVID-19 hospitalization according to ethnicity [7,8]. However, little evidence is available from other countries on this matter. 

Immigration, especially from Latin America, has changed the demographic landscape of Spain over the last two decades. The number of persons living in Spain on 1 January 2021 was 47.40 million, of whom 1.46 million were from Central and South America, 0.81 million were from North Africa, 0.43 million were from Asia, and 0.25 million were from Sub-Saharan Africa [9]. This influx of immigrants into Spain makes it necessary to study whether there are differences in the clinical progress of patients with COVID-19 according to their ethnicity. Few studies have analyzed the effect of ethnicity on patient progress in Spain [10,11,12,13,14,15] or other European Union countries [16]. Although the Spanish healthcare system is universal and accessibility is theoretically guaranteed to all [17,18], there is little information about ethnicity in health outcomes among patients in Spanish hospitals during the COVID-19 pandemic. 

This work analyzes differences according to ethnicity in the adverse outcomes of in-hospital mortality (IHM), admission to the intensive care unit (ICU), the use of invasive mechanical ventilation (IMV), and a composite variable of the three using data from the SEMI-COVID-19 Registry, a large, geographically-diverse surveillance network for COVID-19-associated hospitalizations in 150 Spanish hospitals [19,20].

## 2. Materials and Methods

### 2.1. Study Design and Population

This retrospective cohort study analyzed a nationwide, multicenter registry of patients hospitalized with confirmed COVID-19 in Spain from 1 March 2020 to 30 April 2021. The patients were analyzed according to the pandemic wave in which they were admitted during the study period: the first wave or the second wave. The first wave included patients admitted from 1 March to 30 June 2020 and the second wave included patients admitted from 1 July 2020 to 31 January 2021. 

### 2.2. Definition of Variables

All patient data were obtained from the Spanish Society of Internal Medicine’s SEMI-COVID-19 Registry, in which 150 Spanish hospitals participate. The SEMI-COVID-19 Registry prospectively compiles data on the index admission of patients ≥18 years of age with COVID-19 confirmed microbiologically through a reverse transcription polymerase chain reaction (RT-PCR) or antigen test. More in-depth information about the justification, objectives, methodology, and preliminary results of the SEMI-COVID-19 Registry have recently been published [19,20].

The data on ethnicity were determined based on the place of birth indicated on the patient’s electronic medical record (EMR). For this analysis, ethnicity was categorized as follows: Latin Americans: individuals born in Central and South AmericaAsian: individuals born in AsiaSub-Saharan Africans: individuals born in Sub-Saharan AfricaNorth Africans: individuals born in North African or Middle Eastern countries excluding IsraelEuropean: individuals born in Europe and North America.

The degree of dependence was evaluated using the Barthel Index. Comorbidities were evaluated by means of the age-adjusted Charlson Comorbidity Index (CCI) [21]. Patients were classified as having dyslipidemia, diabetes mellitus, or hypertension if they had a previous diagnosis on their EMR or received pharmacological treatment for these conditions. Atherosclerotic cardiovascular disease was defined as a medical history of coronary heart disease (myocardial infarction, acute coronary syndrome, angina, or coronary revascularization), cerebrovascular disease (stroke, transient ischemic attack), or peripheral arterial disease (intermittent claudication, revascularization, lower limb amputation, or abdominal aortic aneurysm). Chronic pulmonary disease was defined as diagnosis of asthma and/or chronic obstructive pulmonary disease. Malignancy encompassed hematologic neoplasia and/or solid tumors (excluding non-melanoma skin cancer). All baseline comorbidities were gathered from EMR obtained from the hospitals.

The endpoints of the study were 30-day all-cause IHM; admission to the ICU; use of IMV; and a composite variable of IHM, admission to the ICU, and use of IMV.

### 2.3. Statistical Analysis 

Each group’s characteristics were analyzed using descriptive statistics. Continuous and categorical variables were expressed as medians and interquartile ranges (IQR) and as absolute values and percentages, respectively. The differences between groups were analyzed using the Mann–Whitney U test and Kruskal–Wallis test for continuous variables, and Pearson’s chi-square test for categorical variables. Statistical significance was defined as *p* < 0.05.

Differences in IHM, ICU admission, use of IMV, and composite outcome were adjusted by age (0–44, 45–64, or 65+ years), sex, COVID-19 wave, nosocomial acquisition, and baseline CCI category (0–2, 3–4, 5+) using a multivariable regression analysis and a stepwise regression with a threshold of *p* < 0.10. These data are shown in Appendix A. The values are expressed as adjusted odds ratios (OR) with 95% confidence intervals (CI). Statistical analyses were performed using IBM SPSS Statistics v25 (Armonk, NY, USA).

### 2.4. Ethical Aspects

This work was approved by the Institutional Research Ethics Committee of Málaga on 27 March 2020 (Ethics Committee code: SEMI-COVID-19 27-03-20), as per the guidelines of the Spanish Agency of Medicines and Medical Products. All patients gave informed consent. All data collected, processed, and analyzed in this work were anonymized and used only for the purposes of this project. All data were protected in accordance with the Regulation (EU) 2016/679 of the European Parliament and of the Council of 27 April 2016 on the protection of natural persons with regard to the processing of personal data and on the free movement of such data. This study was approved by the Institutional Research Ethics Committees of each participating hospital. The STROBE statement guidelines were adhered to in the execution and reporting of the study. 

## 3. Results

### 3.1. Study Sample

Of the 23,254 hospitalized patients included in the SEMI-COVID-19 Registry, data on ethnicity were not available for 301. Therefore, the total population analyzed included 23,953 patients (median age 69.5 years, 42.9% women). Of them, 7.0% were Latin American, 1.2% were North African, 0.5% were Asian, 0.5% were Sub-Saharan African, and 89.7% were European (Table 1). The patients in the ethnic minority groups were significantly younger than European patients (median (IQR) age range 49.1 (40.5–58.9) to 57.1 (44.1–67.1) years vs. 71.5 (59.5–81.4) years, *p* < 0.001 for all pairwise comparisons of ethnic minority groups vs. European patients). There were significantly more women in the Latin American group (46.1%) and significantly fewer women in the Sub-Saharan African (38.7%), Asian (33.6%), and North African (33.6%) groups than in the European group (42.3%) (*p* < 0.001). Most of the patients were admitted in the first wave (73.5%); there were significantly fewer patients from the first wave in the North African (58.0%) and Sub-Saharan African (58.9%) than the Asian (69.1%), European (73.8%), and Latin American (74.8%) groups (*p* < 0.001).

The patients in ethnic minority groups had significantly fewer comorbidities as assessed by CCI than European patients (median (IQR) range 1 (0–2) to 2 (0–3.0) vs. 4 (2–5) European patients, *p* < 0.001) (Table 1). In general, the ethnic minority groups had lower rates of hypertension, dyslipidemia, non-atherosclerotic heart disease, atherosclerotic cardiovascular disease, diabetes mellitus, malignancy, dementia, and kidney disease than the European patients (Table 1).

### 3.2. Study Outcomes

#### 3.2.1. In-Hospital Mortality

The unadjusted proportion of deaths during hospitalization differed across the ethnic groups, with higher IHM observed among European patients (21.6%) and lower IHM observed among Sub-Saharan African (3.2%), Latin American (7.1%), Asian (10.9%) and North African patients (11.4%) (Table 2; overall *p* < 0.001). After adjusting for demographic differences and CCI, Sub-Saharan African patients were 70% less likely to die during hospitalization than European patients (OR (95% CI) 0.28 (0.10–0.79), *p* = 0.017). The differences observed between the other ethnic minority groups compared to European patients were not statistically significant (Table 3, *p* > 0.05 for all). The results of IHM during the first wave and second wave data are shown in Appendix A.

#### 3.2.2. Admission to the ICU

The unadjusted proportion of patients admitted to the ICU differed across ethnic groups, with higher ICU admission rates observed among North African (16.8%), Asian (14.5%), and Latin American patients (10.3%), and lower ICU admission rates observed among Sub-Saharan African (10.3%) and European patients (9.1%) (Table 2; global *p* < 0.001). After adjusting for demographic differences and baseline clinical characteristics, Latin American and North African patients were 37% and 74% more likely to be admitted to the ICU than European patients (OR (95% CI) 1.37 (1.17–1.60), *p* < 0.001, and OR (95% CI) 1.74 (1.26–2.41), *p* = 0.001) (Table 3), while the differences between Sub-Saharan African and Asian patients compared to European patients were not statistically significant (Table 3, *p* > 0.05 for both). The results of admission to the ICU during the first wave and second wave data are shown in Appendix A.

#### 3.2.3. IMV

The unadjusted proportion of need for IMV differed across ethnic groups, with greater use of IMV observed among North African (10.8%) and Latin American (10.1%) patients and less use of IMV among European (7.0%), Sub-Saharan African (7.3%), and Asian (9.1%) patients (*p* < 0.001). After adjusting for demographic differences, Latin American patients were 43% more likely than European patients to use IMV (OR (95% CI) 1.43 (1.21–1.71), *p* < 0.001) (Table 3), while the differences between North African, Sub-Saharan African, and Asian patients compared to European patients were not statistically significant (Table 3, *p* > 0.05 for all). The results of IMV during the first wave and second wave data are shown in Appendix A.

#### 3.2.4. Composite Outcome

The unadjusted proportion of the composite outcome differed across ethnic groups. It was less common among Latin American (15.8%) and Sub-Saharan African (13.8%) patients and more common among European (27.2%), North African (21.9%), and Asian (21.8%) patients (*p* < 0.001). However, there were no differences among ethnic groups after adjusting for demographic differences (Table 3). The results of the composite outcome during the first wave and second wave data are shown in Appendix A.

#### 3.2.5. Other Outcomes

The median length of hospital stay for the total population was 9 days (IQR:5–14), with significant differences among the ethnic groups (Table 2, total *p* = 0.032). The median length of hospital stay among Latin Americans was significantly shorter than among Europeans (8 days (5–13) vs. 9 days (6–14); *p* = 0.024). The results of other outcomes during all periods of study, and the first wave and second wave data, are shown in Table 2 and Appendix A.

## 4. Discussion

The main finding of this study is that the ethnicity of patients hospitalized with COVID-19 in Spain was not a determining factor of mortality. In this large, nationwide, retrospective study of nearly 24,000 patients hospitalized in Spain with confirmed COVID-19, around 10% of the patients were ethnic minorities, with Latin Americans being the predominant group. These epidemiological data are similar to other Spanish series of patients with COVID-19 [10,11,12,13,14,15,17]. 

As was expected, patients belonging to an ethnic minority group were significantly younger and had fewer comorbidities than the European patients, the group that includes the native Spanish population. This is likely due to the fact that most migrants are of working age and have settled in Spain in the last two decades [22,23].

This study also found that members of ethnic minority groups also had lower non-adjusted rates of mortality and adverse clinical outcomes than European patients. The more favorable COVID-19 outcomes observed in migrants could be justified by the fact that they are a younger, healthier population and it is well known that older age and comorbidities are among the most important risk factors for COVID-19-related complications and mortality [19,22]. However, even after adjusting for age and comorbidities, patients who belonged to ethnic minority groups were found not to have worse outcomes than European patients. These results are different to previous studies in populations in the USA [3,4,5,7,24,25,26] and the UK [27], where the racial and ethnic minority groups experienced disproportionately severe COVID-19-associated outcomes. A systematic review that analyzed ethnic disparities in COVID-19 in the USA concluded that African American/Black and Hispanic populations had higher rates of COVID-19-related mortality, but similar case fatality rates [2]. In another systematic review that included 52 studies (75.0% from the USA, 18.1% from the UK, and 6.9% from Brazil), the COVID-19 mortality rate was significantly higher among Blacks and Hispanics compared to Whites [28]. 

It has been hypothesized that this worse COVID-19 prognosis observed in ethnic minority populations is likely multifactorial [3]. First, some comorbidities such as hypertension, obesity, and chronic kidney disease, which are linked to worse clinical outcomes in COVID-19, are more prevalent in these populations [4,5,29,30,31]. Second, the differences in immune response and angiotensin-converting enzyme 2 expression and/or polymorphism observed among individuals of different ethnicities [32,33], as well as genetic variations that may lead to greater disease severity among individuals of certain ethnicities, have been proposed [34,35,36]. Finally, and probably most importantly, social vulnerability and social determinants of health (poverty, low education, and economic levels) are related to an increased risk of both infection and fatality in COVID-19 [37,38,39]. Additionally, inequities in access to healthcare must be considered as an underlying factor of the poor health outcomes observed [3].

The effect of different healthcare system models on clinical outcomes in COVID-19 has not been well studied. Spain has been ranked among the top countries in the world in regard to healthcare access and quality [40]. Spain’s universal public healthcare system guarantees equal access to all, including immigrants, regardless of their legal status [18,41]. Our results suggest that this universal public healthcare system was critical in minimizing the social determinants of health, palliating the socioeconomic inequalities and lack of access to healthcare that have been linked to ethnic disparities in rates of severe COVID-19 described in other countries [3,29]. 

The impact of COVID-19 on ethnic minority groups in Spain has been analyzed in a few previous studies. In a population-based study of a cohort of adult patients with PCR-confirmed COVID-19 conducted in Madrid, migrants from Sub-Saharan Africa and Latin America showed an increased risk for COVID-19 compared to Spaniards [11]. This higher risk of COVID-19 among minorities have been widely reported in other countries [2,28] and can likely be explained by socioeconomic determinants of health, as discussed previously. 

Regarding the clinical outcomes of COVID-19 in immigrant populations in Spain, the results are not consistent. Several small and/or single-center studies have shown an increased risk of ICU admission and IHM in individuals born in Latin American countries, but the model was not adjusted for comorbidities in these studies [10,12,13]. In another multicenter study from 18 Spanish hospitals that included 10,100 patients hospitalized for COVID-19 during the first wave of infections, 14.8% of whom were not born in Spain, Latin American patients also had higher rates of ICU admission and IHM, but, yet again, no adjustment for comorbidities or clinical and treatment variables was performed [15]. In contrast, a single-center study conducted during the first wave of the pandemic in Madrid found a lower rate of IHM among migrants (73.5% from Latin American countries) after adjusting for comorbidities. Finally, findings similar to our study were reported in a multicenter study in four hospitals in Madrid and Barcelona, which included 5235 patients hospitalized with COVID-19 during the first wave. In that work, Latin American patients had a higher rate of ICU admission, but the IHM rate was not higher; however, comorbidities were not included in the multivariable analysis [14].

Interestingly, the Sub-Saharan African patients in our series showed a lower adjusted mortality rate than other ethnic groups. This good prognosis observed among Sub-Saharan Africans could have several hypothetical explanations. On the one hand, Black Africans, who are assumed to form the majority of the Sub-Saharan African group, have been found to be less likely to carry a gene cluster on chromosome 3 that has been associated with severe COVID-19 [36]. On the other hand, Black patients have also been found to have a lower proportion of abdominal adiposity than Latin American and White patients, and visceral fat has been related to a worse prognosis in COVID-19 [42].

Latin American and North African patients had higher adjusted rates of ICU admission with no associated increase in IHM in our study. The increased frequency of ICU admission in non-European patients could be a result of the younger age of this population, as well as the shortage of resources during the peak of the pandemic’s first wave, which is the period when most patients were included in this study. On the other hand, the high proportion of visceral fat and metabolic syndrome in Latin American [42,43] and North African patients could explain their worse outcomes [44]. Further studies investigating potential metabolic, immunological, and genetic markers in patients of different geographical origins are required to understand the determinants of COVID-19 clinical outcomes.

Our study is the largest series of hospitalized patients with COVID-19 in Spain. However, this work has several limitations. First, it has a retrospective cohort register. Second, most of our patients were included during the first wave of the pandemic. Third, there was a relatively low proportion of patients from Sub-Saharan Africa and Asia. Fourth, in Spain, like most European Union Member States, anti-discrimination laws based on the EU Data Protection Directive and Article 21 of the EU Charter do not permit data collection on the basis of racial and ethnic origin [45]. Paradoxically, this policy may limit the generation of evidence that could contribute to reducing disparities among ethnic groups. In our study, a patient’s ethnicity was approximated according to their country of birth. Finally, our conclusions may not be applicable to other settings with different healthcare system models.

## 5. Conclusions

In Spain, which has a universal public healthcare system, no differences in mortality were observed among minority ethnic groups despite socioeconomic disparities. More well-designed, longitudinal studies are needed to analyze the potential disparities in COVID-19 clinical outcomes among ethnic minority populations in order to identify factors that contribute to them, including the healthcare system model.

## Figures and Tables

**Table 1 jcm-11-01949-t001:** Demographic and clinical characteristics of patients with confirmed COVID-19 by ethnic group.

	M	Total (*n* = 22,953)	Latin Americans (*n* = 1839; 8.0%)	North Africans (*n* = 281; 1.2%)	Sub-Saharan Africans (*n* = 124; 0.5%)	Asians (*n* = 110; 0.5%)	Europeans (*n* = 20,599; 89.7%)	Global *p*	LA vs. E *p*	NA vs. E *p*	SS vs. E *p*	Asian vs. E *p*
Age, years, median (IQR)	0	69.5 (56.7–80.1)	49.1 (40.5–58.9)	57.1 (44.1–67.1)	50.0 (42.3–61.3)	49.7 (41.1–62.4)	71.5 (59.5–81.4)	<0.001	<0.001	<0.001	<0.001	<0.001
Age, years, *n* (%)												
18–39	0	1340 (5.8)	433 (23.5)	46 (16.4)	23 (18.5)	23 (20.9)	815 (4.9)	<0.001	<0.001	<0.001	<0.001	<0.001
40–64	0	8054 (35.1)	1151 (62.6)	150 (53.4)	73 (58.9)	65 (59.1)	6617 (32.1)	<0.001	<0.001	<0.001	<0.001	<0.001
65+	0	13,558 (59.1)	255 (13.9)	85 (30.2)	28 (22.6)	22 (20.0)	13,168 (63.9)	<0.001	<0.001	<0.001	<0.001	<0.001
Sex, female, *n* (%)	0	9740 (42.4)	847 (46.1)	37 (33,6)	48 (38.7)	37 (33.6)	8716 (42.3)	<0.001	0.002	0.001	0.412	0.066
Wave, first, *n* (%)	0	16,879 (73.5)	1375 (74.8)	163 (58.0)	73 (58.9)	76 (69.1)	15,192 (73.8)	<0.001	0.341	<0.001	<0.001	0.268
Nosocomial infection, *n* (%)	0	1088 (4.7)	44 (2.4)	10 (3.6)	5 (4.0)	4 (3.6)	1025 (5.0)	<0.001	<0.001	0.275	0.640	0.517
Dependence, mild and severe, *n* (%)	218	1711 (7.5)	22 (1.2)	1 (0.4)	2 (1.7)	2 (1.8)	1684 (8.3)	<0.001	<0.001	<0.001	<0.001	<0.001
Baseline CCI, median (IQR)	0	3 (2–5)	1 (0–2)	2 (0.0–3.0)	1 (0–3)	1.5 (0–2.5)	4 (2–5)	<0.001	<0.001	<0.001	<0.001	<0.001
Baseline CCI, *n* (%)												
0–2	0	8595 (38.2)	1468 (81.5)	177 (64.4)	82 (67.8)	81 (75.0)	6787 (33.6)	<0.001	<0.001	<0.001	<0.001	<0.001
3–4	0	6328 (28.1)	208 (81.5)	60 (21.8)	25 (20.7)	16 (14.8)	6019 (39.8)	<0.001	<0.001	<0.001	<0.001	<0.001
5+	0	7601 (33.7)	126 (7.0)	38 (13.8)	14 (11.6)	11 (10.2)	7412 (36.7)	<0.001	<0.001	<0.001	<0.001	<0.001
Comorbidities, *n* (%)												
Hypertension	25	11,997 (52.3)	402 (21.9)	101 (35.9)	49 (39.5)	45 (31.3)	11,997 (52.3)	<0.001	<0.001	<0.001	<0.001	0.003
Dyslipidemia	39	9091 (39.7)	389 (21.2)	89 (31.7)	30 (24.2)	25 (22.9)	8558 (41.6)	<0.001	<0.001	0.001	<0.001	<0.001
Non-atherosclerotic heart disease ^a^	54	3423 (14.9)	46 (2.5)	18 (6.4)	6 (4.8)	5 (4.6)	3348 (16.3)	<0.001	<0.001	<0.001	0.001	0.001
Atherosclerotic cardiovascular disease ^b^	81	3895 (17.0)	101 (5.5)	32 (11.5)	7 (5.7)	12 (11.0)	3743 (18.2)	<0.001	<0.001	0.004	<0.001	0.051
Diabetes mellitus	49	4782 (20.9)	212 (11.6)	73 (26.0)	29 (23.8)	19 (17.4)	4449 (21.6)	<0.001	<0.001	0.080	0.529	0.287
Obesity	1806	4775 (22.6)	469 (27.6)	60 (22.4)	28 (22.8)	10 (9.7)	4208 (22.2)	<0.001	<0.001	0.942	0.881	0.002
Chronic pulmonary disease ^c^	53	3626 (15.8)	175 (9.6)	36 (12.8)	19 (15.3)	11 (10.1)	3385 (16.5)	<0.001	<0.001	0.100	0.732	0.073
Malignancy ^d^	33	2290 (10.0)	72 (3.9)	13 (4.6)	8 (6.5)	7 (6.4)	2190 (10.7)	<0.001	<0.001	0.001	0.136	0.152
Dementia	39	2777 (9.9)	24 (1.3)	2 (0.7)	2 (1.6)	1 (0.9)	2248 (10.9)	<0.001	<0.001	<0.001	0.001	0.001
Chronic kidney disease ^e^	37	1481 (6.2)	1358 (6.6)	7 (2.5)	41 (2.2)	5 (4.6)	1358 (6.6)	<0.001	<0.001	0.006	0.430	0.398
HIV	64	122 (0.5)	23 (1.3)	2 (0.7)	6 (4.8)	0 (0.0)	91 (0.4)	<0.001	<0.001	0.499	<0.001	0.486
Obstructive sleep apnea	114	1363 (6.0)	51 (2.8)	11 (3.9)	6 (4.9)	3 (2.8)	11 (3.9)	<0.001	<0.001	0.106	0.530	0.128
Chronic liver disease	60	808 (3.5)	60 (3.3)	15 (5.4)	2 (1.6)	8 (7.4)	723 (3.5)	0.058	0.635	0.098	0.250	0.029
Baseline medication use, *n* (%)												
ASA	110	3726 (16.3)	78 (4.2)	33 (11.8)	11 (8.9)	10 (9.3)	3594 (17.5)	<0.001	<0.001	0.012	0.011	0.024
Statin	99	7317 (32.0)	248 (13.5)	20 (18.5)	25 (20.2)	248 (13.5)	71 (25.3)	<0.001	<0.001	0.002	0.001	0.001
ACE inhibitor	88	3998 (17.5)	152 (8.3)	44 (15.7)	16 (12.9)	18 (16.7)	3768 (18.4)	<0.001	<0.001	0.244	0.117	0.649
ARB	86	4617 (20.2)	148 (8.1)	31 (11.1)	20 (16.3)	8 (7.4)	4410 (21.5)	<0.001	<0.001	<0.001	0.159	0.002
Anticoagulant	78	2565 (11.2)	36 (2.0)	15 (5.4)	6 (4.8)	4 (3.7)	2504 (12.2)	<0.001	<0.001	<0.001	0.012	0.007

*n* (%): number of cases (percentage); M: missing value; IQR: interquartile range; CCI: Charlson Comorbidity Index; E: Europeans; LA: Latino-Americans; NA: North Africans; SS: Sub-Saharans. HIV: human immunodeficiency virus. ASA: acetylsalicylic acid. ACE: angiotensin-converting enzyme. ARB: angiotensin receptor blockers. ^a^ Non-atherosclerotic heart disease includes atrial fibrillation and/or heart failure. ^b^ Atherosclerotic cardiovascular disease includes coronary, cerebrovascular, and/or peripheral vascular disease. ^c^ Chronic pulmonary disease includes chronic obstructive pulmonary diseases and/or asthma. ^d^ Malignancy includes solid tumors or hematological neoplasia. ^e^ Chronic kidney disease is defined as an estimated glomerular filtration rate (eGFR) <45 mL/min/1.73 m^2^ according to the CKD-EPI equation.

**Table 2 jcm-11-01949-t002:** Outcomes of patients with confirmed COVID-19 by ethnic group.

	M	Total (*n* = 22,953) ^ϕ^	Latin Americans (*n* = 1839; 8.0%) ^ϕ^	North Africans (*n* = 281; 1.2%) ^ϕ^	Sub-Saharan Africans (*n* = 124; 0.5%) ^ϕ^	Asians (*n* = 110; 0.5%) ^ϕ^	Europeans (*n* = 20,599; 89.7%) ^ϕ^	Global *p*	LA vs. E *p*	NA vs. E *p*	SS vs. E *p*	Asian vs. E *p*
Main outcomes, *n* (%)												
In-hospital mortality	0	4636 (20.2)	130 (7.1)	32 (11.4)	4 (3.2)	12 (10.9)	4458 (21.6)	<0.001	<0.001	<0.001	<0.001	0.006
Intensive care unit admission	11	2186 (9.5)	239(13.0)	47 (16.8)	13 (10.3)	16 (14.5)	1871 (9.1)	<0.001	<0.001	<0.001	0.590	0.047
Invasive mechanical ventilation	92	1664 (7.3)	185 (10.1)	30 (10.8)	9 (7.3)	10 (9.1)	1430 (7.0)	<0.001	<0.001	0.014	0.880	0.384
Composite outcome	57	5983 (26.1)	289 (15.8)	61 (21.9)	17 (13.8)	24 (21.8)	5592 (27.2)	<0.001	0.001	0.050	0.001	0.205
Other outcomes, *n* (%)												
Non-invasive mechanical ventilation	100	1325 (5.8)	113 (6.2)	24 (8.6)	9 (7.3)	7 (6.4)	1172 (5.7)	0.255	0.421	0.040	0.443	0.770
High-flow oxygen therapy	158	2199 (9.6)	215 (11.8)	45 (16.2)	16 (13)	11 (10.0)	1912 (9.3)	<0.001	0.001	<0.001	0.164	0.814
Length of hospital stay, days, median (IQR)	145	9 (5–14)	8 (5–13)	9 (6–15)	9 (6–15)	8 (5–16)	9 (6–14)	0.023	0.027	0.104	0.297	0.832
Length of hospital stay >10 days, *n* (%)	145	10,276 (45.1)	762 (41.5)	137 (49.3)	61 (49.5)	48 (43.6)	9268 (45.3)	0.012	0.002	0.186	0.341	0.772

^ϕ^ All patients; M: missing value; E: Europeans; LA: Latin Americans; NA: North Africans; SS: Sub-Saharan Africans; IQR: Interquatile range.

**Table 3 jcm-11-01949-t003:** Crude and multivariable logistic regression model for mortality, intensive care unit admission, and invasive mechanical ventilation among patients with confirmed COVID-19 by ethnicity.

	Crude		Adjusted Model ^a^	
Independent Variables	Odds Ratio (95% CI)	*p*	Odds Ratio (95% CI)	*p*
In-hospital mortality				
Latin Americans	0.27 (0.23–0.33)	<0.001	1.11 (0.90–1.36)	0.320
North Africans	0.46 (0.32–0.67)	<0.001	1.05 (0.70–1.60)	0.786
Sub-Saharan Africans	0.12 (0.04–0.32)	<0.001	0.28 (0.10–0.79)	0.017
Asians	0.44 (0.24–0.80)	0.008	1.50 (0.77–2.93)	0.320
Europeans	Ref		Ref	
Intensive care unit admission				
Latin Americans	1.49 (1.29–1.72)	<0.001	1.37 (1.17–1.60)	<0.001
North Africans	2.01 (1.47–2.71)	<0.001	1.74 (1.26–2.41)	0.001
Sub-Saharan Africans	1.17 (0.65–2.08)	0.590	1.03 (0.57–1.86)	0.903
Asians	1.70 (1.00–2.90)	0.050	1.49 (0.87–2.56)	0.145
Europeans	Ref		Ref	
Invasive mechanical ventilation				
Latin Americans	1.50 (1.27–1.76)	<0.001	1.43 (1.21–1.71)	<0.001
North Africans	1.60 (1.10–2.36)	0.015	1.50 (1.01–2.21)	0.051
Sub-Saharan Africans	1.05 (0.53–2.08)	0.880	1.01 (0.50–2.01)	0.974
Asians	1.33 (0.68–2.56)	1.335	1.23 (0,63–2.38)	0.534
Europeans	Ref		Ref.	
Composite outcome				
Latin Americans	0.50 (0.44–0.57)	<0.001	1.13 (0.98–1.31)	0.082
North Africans	0.75 (0.57–1.00)	0.051	1.27 (0.93–1.72)	0.122
Sub-Saharan Africans	0.43 (0.26–0.71)	0.001	0.79 (0.46–1.35)	0.398
Asians	0.75 (0.47–1.17)	0.207	1.521 0.94–2.44)	0.085
Europeans	Ref		Ref	

Ref.: reference. ^a^ Adjusted for age group, sex, wave, place of acquisition, and baseline Charlson Comorbidity Index group. See Appendix A for full models.

## Data Availability

J.-M.R.-R. and R.G.-H. have full access to the data and are the guarantors for the data.

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
