# Peer review of "Ethnicity and Clinical Outcomes in Patients Hospitalized for COVID-19 in Spain: Results from the Multicenter SEMI-COVID-19 Registry"

_jcm, 2022, doi:10.3390/jcm11071949_

Round 1

Reviewer 1 Report

Authors have studied clinical outcomes in hospitalized for COVID-19 .The paper presents new and interesting insights. The results are presented in a straightforward manner and might draw the reader’s attention. Therefore, I would suggest a few minor comments.

  1. For betterment of introduction part, authors are suggested to refer 106, 1197–1211(2021). 106, 1213–1227 (2021.)
  2. The conclusion part is unclear and must be precise for understanding what authors have achieved through this study.

Author Response

Author's Reply to the Review Report (Reviewer 1)

Authors have studied clinical outcomes in hospitalized for COVID-19 .The paper presents new and interesting insights. The results are presented in a straightforward manner and might draw the reader’s attention. Therefore, I would suggest a few minor comments.

  1. For betterment of introduction part, authors are suggested to refer 106, 1197–1211(2021). 106, 1213–1227 (2021.

Reply. Dear reviewer, you recommend that we use these two references for inclusion in the introduction. We have not been able to refer 106, 1197–1211(2021). 106, 1213–1227 (2021.because we do not have the title of the journal to know which reference we could include in the introduction.

  1. The conclusion part is unclear and must be precise for understanding what authors have achieved through this study.

Reply. We thank the reviewer for his suggestion regarding the conclusions. We have re-write the conclusion to being precise about we have achieved through this study

Reviewer 2 Report

The authors tried to study the effect of ethnicity on clinical outcome.

1. The difference of the proportions of each ethnicity is too large. For example, the ratio of sub- Saharan Africans is just 0.5% and the Europeans is 89.7%. The huge proportion difference will lead the huge bias. The amount of cases of each ethnicity included the study should be balanced.

2. In the study, the ethnicity was defined by the place of birth. The reviewer thinks that this is not rigorous. For example, we could not define the ethnicity the second generation immigrants by the place of birth. There are more faults are even in the definition of the place of birth. As is well-known, nearly twenty Middle East countries are Belong to Asia, like Iraq, UAE, Saudi Arabia…etc. But the authors defined the whole Middle East countries belong to North Africans. It is also well known that the individuals born in North America included Immigrants of many kinds of the ethnicity. But the authors defined all the individuals born in North America belong to Europeans.

It is suggested that the authors correct the  the definition of ethnicity by the common sense and balance the case amount of each ethnicity and re-analysis the data.

Author Response

Manuscript ID jcm-1624478

Type Article

Title Ethnicity and clinical outcomes in patients hospitalized for COVID-19 in Spain: Results from the multicenter SEMI-COVID-19 registry

Author's Reply to the Review Report (Reviewer 2)

The authors tried to study the effect of ethnicity on clinical outcome.

  1. The difference of the proportions of each ethnicity is too large. For example, the ratio of sub- Saharan Africans is just 0.5% and the Europeans is 89.7%. The huge proportion difference will lead the huge bias. The amount of cases of each ethnicity included the study should be balanced.

Reply. We appreciate the reviewer's comments, although the number of people with COVID-19 included in the registry cannot be modified, the number of patients included in each ethnic group is what it is and cannot be increased.

In Spain, the main ethnic group is from Latin America and this is reflected in those admitted to the hospital for COVID-19. 

  1. In the study, the ethnicity was defined by the place of birth. The reviewer thinks that this is not rigorous. For example, we could not define the ethnicity the second generation immigrants by the place of birth. There are more faults are even in the definition of the place of birth. As is well-known, nearly twenty Middle East countries are Belong to Asia, like Iraq, UAE, Saudi Arabia…etc. But the authors defined the whole Middle East countries belong to North Africans. It is also well known that the individuals born in North America included Immigrants of many kinds of the ethnicity. But the authors defined all the individuals born in North America belong to Europeans.

It is suggested that the authors correct the definition of ethnicity by the common sense and balance the case amount of each ethnicity and re-analysis the data

Reply: we appreciate the reviewer's suggestion.  We define the ethnic group by geographic area of birth of the patient.

We have to say that immigration is a relatively recent phenomenon since the late 1990s, especially since 1997. Second-generation immigrants are still uncommon in those over 18 years of age who are included in the cohort. In Spain, the children of immigrants born abroad, the majority of whom were born in Spain, are Spanish and the country of birth of their father is not known.

We regret to comment to the reviewer that we cannot re-define patients with COVID-19 not born in Spain.

We have used this grouping of countries, since the profile of these patients has a denominator which is the Arabic people. Moreover, this geographic classification has been used in other epidemiological studies by geographic areas. This the case of geographic clustering in patients living with HIV, where a well-defined geographic area is as analyze the participation of in HIV research  UNAIDS (2018), as Middle East and North Africa (ME&NA). (https://www.unaids.org/sites/default/files/media_asset/unaids-data-2018_en.pdf)

We understand that patients from North America have been considered as Europeans, and maybe some of them were North Americans but of Latin American or Asian origin. We recognize this limitation, but it must be said that very few patients from North America were ingested.

Reviewer 3 Report

The manuscript jcm-1624478m entitled “Ethnicity and clinical outcomes in patients hospitalized for COVID-19 in Spain: Results from the multicenter SEMI-COVID-19 registry” by Jose-Manuel Ramos-Rincon and co-workers aims to analyze clinical outcomes according to ethnic groups in patients hospitalized for COVID-19 in Spain.

The study analyzed hospitalized patients with confirmed COVID-19 in 150 Spanish hospitals (SEMI-COVID-19 Registry) from March 1, 2020 to December 31, 2021. Clinical outcomes were assessed according to ethnicity (Latin Americans, Sub-Saharan Africans, Asians, North Africans, Europeans). The outcomes were in-hospital mortality (IHM), intensive care unit (ICU) admission, and use of invasive mechanical ventilation (IMV). Associations be-tween ethnic groups and clinical outcomes adjusted for patient characteristics and baseline Charlson Comorbidity Index values and wave were evaluated using logistic regression. Of 23,953 patients (median age 69.5 years, 42.9% women), 7.0% were Latin American, 1.2% were North African, 0.5% were Asian, 0.5% were Sub-Saharan African, and 89.7% were European. Ethnic minority patients were significantly younger than European patients. The unadjusted IHM was higher in European (21.6%) versus North African (11.4%), Asian (10.9%), Latin American (7.1%), and Sub-Saharan African (3.2%) patients. After further adjustment, IHM was lower in Sub-Saharan African (OR 0.28 [0.10-0.79], p=0.017) versus European patients while ICU admission rates were higher in Latin American and North African versus European patients (OR [95%CI] 1.37 [1.17-1.60], p<0.001) and (OR [95%CI] 1.74 [1.26-2.41], p<0.001). Moreover, Latin American patients were 39% more likely than European patients to use IMV (OR [95%CI] 1.43 [1.21-1.71], p<0.001). The adjusted IHM was similar in all groups except for Sub-Saharan Africans, who had lower IHM. Latin American patients were admitted to the ICU and required IMV more often.

The researchers concluded that ethnicity Is not a determining factor for mortality.

In other systematic review that analyzed ethnic disparities in COVID-19 in the USA, it was found that African American/Black and Hispanic populations had higher rates of COVID-19-related mortality but. In another systematic review that included 52 studies (75.0% 281 from the USA, 18.1% from the UK, and 6.9% from Brazil), the COVID-19 mortality rate 282 was significantly higher among Blacks and Hispanics compared to Whites.

Differences in access to healthcare among ethnic minorities in other countries may be 356 a determining factor of poor prognosis in COVID-19.

The manuscript is very well written and English language is of quality.

The methodology used is clear, logic and well discussed.

Results are meaningful and significant.

Discussion is balanced and based on the data.

Author Response

Manuscript ID jcm-1624478

Type Article

Title Ethnicity and clinical outcomes in patients hospitalized for COVID-19 in Spain: Results from the multicenter SEMI-COVID-19 registry

Author's Reply to the Review Report (Reviewer 3)

The manuscript jcm-1624478m entitled “Ethnicity and clinical outcomes in patients hospitalized for COVID-19 in Spain: Results from the multicenter SEMI-COVID-19 registry” by Jose-Manuel Ramos-Rincon and co-workers aims to analyze clinical outcomes according to ethnic groups in patients hospitalized for COVID-19 in Spain.

The study analyzed hospitalized patients with confirmed COVID-19 in 150 Spanish hospitals (SEMI-COVID-19 Registry) from March 1, 2020 to December 31, 2021. Clinical outcomes were assessed according to ethnicity (Latin Americans, Sub-Saharan Africans, Asians, North Africans, Europeans). The outcomes were in-hospital mortality (IHM), intensive care unit (ICU) admission, and use of invasive mechanical ventilation (IMV). Associations be-tween ethnic groups and clinical outcomes adjusted for patient characteristics and baseline Charlson Comorbidity Index values and wave were evaluated using logistic regression. Of 23,953 patients (median age 69.5 years, 42.9% women), 7.0% were Latin American, 1.2% were North African, 0.5% were Asian, 0.5% were Sub-Saharan African, and 89.7% were European. Ethnic minority patients were significantly younger than European patients. The unadjusted IHM was higher in European (21.6%) versus North African (11.4%), Asian (10.9%), Latin American (7.1%), and Sub-Saharan African (3.2%) patients. After further adjustment, IHM was lower in Sub-Saharan African (OR 0.28 [0.10-0.79], p=0.017) versus European patients while ICU admission rates were higher in Latin American and North African versus European patients (OR [95%CI] 1.37 [1.17-1.60], p<0.001) and (OR [95%CI] 1.74 [1.26-2.41], p<0.001). Moreover, Latin American patients were 39% more likely than European patients to use IMV (OR [95%CI] 1.43 [1.21-1.71], p<0.001). The adjusted IHM was similar in all groups except for Sub-Saharan Africans, who had lower IHM. Latin American patients were admitted to the ICU and required IMV more often.

The researchers concluded that ethnicity Is not a determining factor for mortality.

In other systematic review that analyzed ethnic disparities in COVID-19 in the USA, it was found that African American/Black and Hispanic populations had higher rates of COVID-19-related mortality but. In another systematic review that included 52 studies (75.0% 281 from the USA, 18.1% from the UK, and 6.9% from Brazil), the COVID-19 mortality rate 282 was significantly higher among Blacks and Hispanics compared to Whites.

Differences in access to healthcare among ethnic minorities in other countries may be 356 a determining factor of poor prognosis in COVID-19.

 Reply: The first sentences are part of one part of the discussion and the second paragraph is part of the first part of our conclusions.

The manuscript is very well written and English language is of quality.

Replay: thanks for your words

The methodology used is clear, logic and well discussed.

Replay: thanks for your words

Results are meaningful and significant.

Replay; thanks for your words

Discussion is balanced and based on the data

Replay; thanks for your words

This manuscript is a resubmission of an earlier submission. The following is a list of the peer review reports and author responses from that submission.